# Novel Antimicrobial Peptides from a Cecropin-Like Region of Heteroscorpine-1 from *Heterometrus laoticus* Venom with Membrane Disruption Activity

**DOI:** 10.3390/molecules26195872

**Published:** 2021-09-28

**Authors:** Rima Erviana, Yutthakan Saengkun, Prapenpuksiri Rungsa, Nisachon Jangpromma, Patcharaporn Tippayawat, Sompong Klaynongsruang, Jureerut Daduang, Sakda Daduang

**Affiliations:** 1Faculty of Pharmaceutical Sciences, Khon Kaen University, Khon Kaen 40002, Thailand; rima@umy.ac.id (R.E.); abuxas.saengkun@gmail.com (Y.S.); prapenpuksiri@gmail.com (P.R.); 2School of Pharmacy, Universitas Muhammadiyah Yogyakarta, Bantul, Yogyakarta 55183, Indonesia; 3Protein and Proteomics Research Center for Commercial and Industrial Purposes (ProCCI), Khon Kaen University, Khon Kaen 40002, Thailand; nisaja@kku.ac.th (N.J.); somkly@kku.ac.th (S.K.); 4Faculty of Associated Medical Sciences, Khon Kaen University, Khon Kaen 40002, Thailand; patchatip@kku.ac.th (P.T.); jurpoo@kku.ac.th (J.D.)

**Keywords:** cecropin, CeHS-1, antimicrobial peptide, sequences modification

## Abstract

The increasing antimicrobial-resistant prevalence has become a severe health problem. It has led to the invention of a new antimicrobial agent such as antimicrobial peptides. Heteroscorpine-1 is an antimicrobial peptide that has the ability to kill many bacterial strains. It consists of 76 amino acid residues with a cecropin-like region in N-terminal and a defensin-like region in the C-terminal. The cecropin-like region from heteroscorpine-1 (CeHS-1) is similar to cecropin B, but it lost its glycine-proline hinge region. The bioinformatics prediction was used to help the designing of mutant peptides. The addition of glycine-proline hinge and positively charged amino acids, the deletion of negatively charged amino acids, and the optimization of the hydrophobicity of the peptide resulted in two mutant peptides, namely, CeHS-1 GP and CeHS-1 GPK. The new mutant peptide showed higher antimicrobial activity than the native peptide without increasing toxicity. The interaction of the peptides with the membrane showed that the peptides were capable of disrupting both the inner and outer bacterial cell membrane. Furthermore, the SEM analysis showed that the peptides created the pore in the bacterial cell membrane resulted in cell membrane disruption. In conclusion, the mutants of CeHS-1 had the potential to develop as novel antimicrobial peptides.

## 1. Introduction

Antimicrobial resistance is one of the greatest threats to the global health problem that has become a challenge for infectious disease treatments [1]. Approximately 700,000 people worldwide are annually killed by antimicrobial resistance, and some researchers estimate that the number may rise to 10 million people by 2050 [2]. It is a complex multifactorial problem that can lead to ineffective therapy, raising the cost of treatment, morbidity, and mortality [3]. Moreover, pandemic situations have resulted in the excessive use of antibiotics which may worsen the threat of antimicrobial resistance [4,5]. The significant increase in the incidence of antimicrobial resistance leads to the development of strategies to discover new antimicrobials that are effective against resistant bacteria [6].

Antimicrobial peptides are some promising novel antibiotics and have great potential to battle bacteria and other microbial [7]. They have been used strategically as the first line of protection against invading pathogens [8]. They are some short stretches of amino acids found in bacteria, fungi, insects, plants, and animals that play an essential role in the innate immune system. Furthermore, antimicrobial peptides have become the main focus due to their potential to combat multidrug-resistant microbes [9]. Another advantage is that they have relatively low minimal inhibitory concentrations for both Gram-positive and Gram-negative bacteria as well as low toxicity against normal cells [10]. Besides, antimicrobial peptides also have a rapid killing effect, with killing ability in seconds upon initial contact with the cell membrane. Moreover, they can enhance the activities of conventional antibiotics through synergistic effects [11]. Some studies of conjugated β-lactam antibiotics with cationic antimicrobial peptides resulted in the significant increasing activity of both drugs [12,13].

Among antimicrobial peptides that have been established, most of them consist of short amino acid chains, containing approximately 5 to 40 amino acid residues [14]. Although there are many heterogeneous amino acid sequences and variations of the secondary structure of the peptides that have been introduced, they are generally cationic, amphipathic, and mainly carry on an α-helix structure. These characteristics will facilitate the peptides to interact with and disrupt the lipid of cell membranes [15,16,17].

Most of the potent antimicrobial peptides have a range of net charges from +2 to +9 [15]. The positive charge of antimicrobial peptides is beneficial for their activity against bacteria as it can selectively kill the bacteria without affecting normal cells. The bacterial cell membranes usually carry more negative charges than the normal mammalian cell membranes [18]. The interaction between the bacterial membrane and antimicrobial peptides is greatly affected by these conditions. When cationic antimicrobial peptides are combined with anionic cell membranes, it leads to an unstable cell membrane. The peptides are later inserted into the lipid bilayer of the cell membrane to result in cell death [19]. In other words, the more the positive charge of the peptides facilitates the attachment of the peptides to the bacterial cell membrane, the higher their activity against the bacteria will be [20].

Moreover, hydrophobicity and amphipathicity are important peptide properties that contribute to the interaction potential between the peptides and the composition of the bacterial cell membrane [16]. Upon the attachment of antimicrobial peptides to the bacterial cell membrane by the electrostatic interaction, hydrophobicity influences the partitioning degree of the peptide into the membrane lipid bilayer [15]. It will facilitate the insertion of the peptides through the membrane resulting in membrane pore formation [21]. The percentages of hydrophobic residues in natural antimicrobial peptides vary from 40% to 60%. This condition is suitable since both hydrophobic and hydrophilic residues in antimicrobial peptides are needed for inserting them into the bacterial membrane [22].

Approximately 5000 recorded antimicrobial peptides have been investigated until now [23]. They vary in length, charge, and structure, contributing to the development of the databases of antimicrobial peptides [22]. Since the 1980s, abundant antimicrobial peptides have been purified from many sources such as plants, animals, and fungi [24]. Furthermore, many efforts have been attempted to develop potent antimicrobial activity by modifying its amino acid sequences. The modification such as substitution, deletion, truncation, and hybridization [20,25,26] have changed the structure of the peptides, which also influence its physicochemical properties [15,22].

In the present study, the researcher attempted to develop the new peptides from a cecropin-like region of Heteroscorpine-1. Heteroscorpine-1 is an antimicrobial peptide purified from the venom of scorpion *Heterometrus laoticus.* It is categorized as a scorpine peptide that originally was purified from *Pandinus imperator* venom [27]. Scorpine peptides are characterized as a cationic antimicrobial peptide that structurally consists of a defensin-like peptide in the C-terminal and a cecropin-like peptide in the N-terminal. The antimicrobial activity of this family associated with the N-terminal sequence similar to antimicrobial peptide cecropin and C-terminal sequence similar to defensin or K^+^ channel blocking peptides [28]. The previous study has been conducted, revealing that Heteroscorpine-1 had strong antimicrobial activities against *Bacillus subtilis, Klebsiella pneumoniae,* and *Pseudomonas aeruginosa* [29]. In this study, the researcher attempted to investigate the N-terminal part of Heteroscorpine-1 as a cecropin-like peptide. Based on the analysis structure, the cecropin-like region of Heteroscorpine-1 (CeHS-1) is similar to cecropin B from *Antheraea pernyi* with the deletion of the Alanine-Glycine-Proline hinge region [30].

Cecropin is a group of antimicrobial peptides that usually contain 34-55 amino acid residues. It is characterized by an N-terminal basic, amphipathic domain linked to a more hydrophobic C-terminal segment. The amphipathic N-terminal domain and hydrophobic C-terminal are connected through a flexible proline-glycine-rich hinge region [31]. Many studies have characterized the antimicrobial activity of cecropin against several Gram-positive and negative bacteria [32]. Based on the explanation above, this study aims to develop the mutant of CeHS-1 with the increasing antimicrobial activity. Specifically, the researcher attempted to optimize the physicochemical properties and analyzed the structure of the mutant peptides.

## 2. Results

### 2.1. Peptides Design and Physicochemical Properties

Initially, the sequence analysis of CeHS-1 was carried out. It compared CeHS-1 with the established potent antimicrobial peptides as the physicochemical descriptor to predict the antimicrobial activity of CeHS-1. Based on the sequence analysis, CeHS-1 had strong similarities with some peptides categorized as cecropin, as shown in Table 1. The comparison between CeHS-1 and cecropin B showed similarities in positioning some hydrophobic and positively charged residues (showed as amino acid sequences with blue color in Table 1). It indicated that CeHS-1 had the potential to be an antimicrobial peptide. Moreover, the sequence analysis also showed that CeHS-1 had hydrophobic residues located on the same surface (underlined amino acid residues).

In the design of mutant peptides, CeHS-1 was used as the template for peptides engineering, while cecropin B was set as an ideal structure of antimicrobial peptides. Two mutants were developed, namely, CeHS-1 GP and CeHS-1 GPK. The amino acid sequences of the mutant peptides are shown in Figure 1, while the secondary structure prediction and the physicochemical properties are shown in Table 2. CeHS-1 GP was designed by deleting glutamic acid and aspartic acid in Positions 5, 6, and 13 of CeHS-1. Furthermore, the modification was also made by adding glycine and proline in positions 22 and 23 of CeHS-1. CeHS-1 GPK was modified from CeHS-1 GP with the substitution of Asparagine to Lysine in Position 19.

The substitution and deletion of amino acids in mutant peptides led to the change in their physicochemical properties. Meanwhile, the addition of glycine-proline hinge region and the deletion of three negatively charged amino acids, glutamic acid, and aspartic acid in positions 5, 6, and 13, changed the charges, % hydrophobic residues, hydrophobicity, and mean hydrophobicity of CeHS-1 GP to +5, 43.24, 0.309, and 0.463, respectively. The substitution of Asparagine residue in position 19 to Lysine increased the net charge from +5 in CeHS-1 GP to +6 in CeHS-1 GPK with other properties similar between CeHS-1 GP and CeHS-1 GPK. The mutant peptides properties adopted the ideal net charges ranging from +4–+6, and hydrophobic residues were between 40–60% [35].

Hereinafter, to confirm the amphipathicity of the peptides, the distribution of hydrophobic and positively charged amino acids was visualized by the helical wheel projection (Figure 2a). Upon the addition of the glycine-proline hinge region and the deletion of negatively charged amino acids, the mutant peptides showed amphipathicity when they adopted the helical structure. In terms of the position, most hydrophobic amino acids are located on one side of the helical structure (amino acids with yellow color), while most of the positively charged amino acids are located on the other side of the helical structure (amino acids with blue color). This position is the ideal structure of antimicrobial peptides [22].

The tridimensional projection confirmed that all peptides adopted an α-helix structure. It also showed the formation of a helical kink in the middle of the helical structure due to the addition of the glycine-proline hinge region (Figure 2b). The tridimensional projections showed that adding the proline-glycine hinge region in CeHS-1 GP and CeHS-1 GPK changed the kink dimension, despite increasing their percentage of helicity.

### 2.2. Structural Analysis of Peptides

To investigate the secondary structure of the peptides, circular dichroism (CD) spectra analysis was performed. The spectra of the peptides were measured in water and membrane mimic conditions in the presence of 40% Trifluoroethanol aqueous solution (TFE). Structural analysis showed that the sequence modification resulted a big impact on the α-helical content of the peptides both in aqueous and membrane mimic environment (Figure 3). In the membrane mimic environment, CeHS-1 GP and CeHS-1 GPK showed two negative peaks at about 208 and 222 nm, demonstrated typical helical structure predisposition [36], while the peeks was not shown in CeHS-1 GP. The deconvolution of spectrum was determined by using K2D2 program [37]. Data in Table 3 show that the α-helix content of CeHS-1 GP and CeHS-1 GPK was similar, 62% in water and increased to 87.59% in 40% TFE indicating the helicity of the mutant peptides was increased in membrane mimic environment. CeHS-1 tend to exhibit random coil structure as it showed only 8.02% of α-helix content in both water and 40% TFE.

### 2.3. Antimicrobial Activity

The antimicrobial activity of peptides was determined by defined minimum inhibitory concentration (MIC) and minimum bactericidal concentration (MBC). Seven bacterial strains were used as the tested microorganism, including Gram-positive *Staphylococcus aureus* ATCC 25923, *Basilus subtilis* TISTR 008, *Pseudomonas aeruginosa* ATCC 27853; Gram-negative *Escherichia coli* ATCC 25922 and *Klebsiella pneumoniae* ATCC 27736; resistant strain *Klebsiella pneumoniae* and *Methicillin Resistant Staphylococcus aureus* (MRSA).

The antimicrobial activity assay showed that CeHS-1, CeHS-1 GP, and CeHS-1 GPK had the ability to inhibit the growth of tested bacteria (Table 4). The mutant peptides, namely, CeHS-1 GP and CeHS-1 GPK, presented a higher activity than CeHS-1, with the MIC ranging between 16–32 μg/mL for CeHS1 GP, and 8–32 μg/mL for CeHS-1 GPK. They denoted good antibacterial activity in both Gram-positive and Gram-negative bacteria. Among the tested peptides, CeHS-1 GPK showed the highest antimicrobial activity, with the lowest MIC. In relation to the MIC, CeHS-1 GPK also showed the lowest MBC compared to other peptides; thus, it had higher activity against *E. coli* and *B. subtilis*. In contrast, the data showed that CeHS-1 only could inhibit the peptides without killing the tested bacteria.

### 2.4. Synergistic Activity Assay

As the peptides with the higher activity, the researcher attempted to investigate the synergistic activity of CeHS-1 GPK with the established conventional antibiotic. The combination was set between CeHS-1 GPK with Ampicillin and Kanamycin. The synergistic effect between two drugs was presented by the fractional inhibitory concentration (FIC) index. As shown in Table 5, the combination of CeHS-1 GPK and Ampicillin resulted in an FIC value of 0.75 in *E. coli*, *B. subtilis*, and *P. aeruginosa*, indicating that the combination resulted in addition. However, the combination in *S. aureus* was indifferent with the FIC value of 3. Meanwhile, in the combination between CeHS-1 GPK and Kanamycin, the collaboration of the drugs resulted in synergism in *B. subtilis,* and *P. aeruginosa* with an FIC value of 0.5. Although the FIC value of the peptide combination with Kanamycin was lower than the combination with Ampicillin in *S. aureus*, the combination still resulted in the indifferent action.

### 2.5. Time-Killing Assay

The time-killing activity of the peptides was assayed against *E. coli* treated with peptides in the concentration 32 μg/mL. The result exhibited that the bacteria treated with CeHS-1 GPK decreased rapidly and were killed in less than 4 h, while those with CeHS-1 GP were killed after 8 h (Figure 4). In contrast, CeHS-1 could inhibit the growth of the bacteria but did not show the ability to kill the bacteria after 12 h of incubation.

### 2.6. Outer Membrane Permeability Assay

The fluorescent probe 1-*N*-phenilnaphthylamine (NPN) was used to identify the ability of the peptides to increase the membrane permeability of the peptides. Normally, NPN cannot effectively cross the bacterial outer membrane. It fluoresces weakly in an aqueous environment. However, in the hydrophobic environment, such as phospholipids, it produces strong fluorescent at an excitation wavelength of 350 nm and an emission wavelength of 420 nm [38]. When the peptides break down the membrane permeability, NPN will easily bind bacterial phospholipids.

In this study, the outer membrane permeability was tested in *E. coli*. When the bacteria were exposed to CeHS-1, CeHS-1 GP, and CeHS-1 GPK, the outer bacterial membrane permeability immediately rose, showed by the increased fluorescent intensity of the sample upon the peptide treatment (Figure 5a). The result denoted that CeHS-1 GPK performed the higher fluorescent intensity, indicating that it had the best ability to disrupt the bacterial outer membrane permeability. Meanwhile, another data compared the ability of peptides to disrupt the membrane permeability in the various concentration of peptides. The data comparison was analyzed at the second minutes after peptides exposure. The result showed that the increased fluorescent intensity correlated to the elevated concentration of peptides (Figure 5b). It was indicated that the higher peptides concentration corresponds to the higher bacterial membrane disruption activity.

### 2.7. Inner Membrane Permeability Assay

The fluorescent probe propidium iodide (PI) was used to determine the effect of the peptide’s treatment towards the bacterial cell membrane of *E. coli* and *S. aureus*. In accordance with the ability to disrupt the outer bacterial membrane permeability, CeHS-1 GPK presented the highest ability to disrupt the inner membrane permeability. While CeHS-1 showed the ability to disrupt the bacterial outer membrane, it was unable to disrupt the bacterial inner membrane, both in *S. aureus* and *E. coli* (Figure 6a and Figure 7a). As shown in Figure 6b and Figure 7b, the activity of peptides in the bacterial inner membrane also influenced by its concentration.

### 2.8. Scanning Electron Microscope (SEM) Analysis

The effect of treating bacteria with the peptides was determined by scanning electron microscope (SEM). The scanning showed that all peptides treatment effect on the bacterial cell surface morphology compared to the control cells. After *E. coli* was treated with half MIC of CeHS-1 for 24 h, the pore was formed (Figure 8b). Furthermore, after treated with CeHS-1 GP and CeHS-1 GPK, membrane integrity of the bacterial cells was changing, some part of the membrane was ripped, and blebs were discernible (Figure 8c,d) (the affected bacterial cell membrane was marked with the red arrows). When the surface of peptide-treated bacteria was seen using SEM, blebs of the cytoplasmic membrane were more easily recognized [39]. Similar with these result in *E. coli*, *S. aureus* treated with peptides, appeared of membrane discontinuity resulting leakage of some cytoplasmic content (Figure 8f–h).

### 2.9. Hemolytic Activity Assay

The three tested peptides presented low hemolytic activity on the human red blood cells (hRBC). Upon testing the highest concentration (200 μg/mL), the percentage of the hemolytic activities was deficient compared to the positive control Triton X-100 (Figure 9). This result indicated that the modification of amino acid sequences did not affect the hemolytic activity of the peptides until the highest tested concentration.

### 2.10. Cytotoxic Assay

The cytotoxic assay was needed to evaluate the safety of the peptides towards normal cells. Antimicrobial peptides should be selective to kill the bacteria without harm to normal cells. In this case, the toxicity of peptides was evaluated using the macrophage (RAW 264.7) cell line and human skin keratinocyte (HaCat) cell line. The result in Figure 10 showed that the structure modification did not make a significant increase in peptides toxicity. Moreover, CeHS-1 GPK showed low toxicity in its MIC. In the higher concentration, the toxicity of the peptides increased, despite the toxicity of the peptides were still acceptable in the highest tested concentration (150 μg/mL).

## 3. Discussion

In this study, the researcher designed new antimicrobial peptides from a cecropin-like region of Heteroscorpine-1, CeHS-1. The strategy concerning the qualitative structure-activity relationship was used to design new potent antimicrobial peptides [22]. This strategy could predict the ideal physicochemical properties of the peptides related to their antimicrobial activity. The ideal physicochemical properties of mutant peptides were optimized based on the template by changing the hydrophobicity, net charge, and secondary structure of the peptides [26]. The modification was performed by deleting the negatively charged amino acids, adding the positively charged amino acids, and analyzing the position of hydrophobic amino acid residues.

The first concept of the design was the addition of the glycine-proline hinge region in the middle of the CeHS-1 sequence. The glycine-proline hinge region was added to the mutant peptides, considering that almost all cecropin and cecropin-like peptides contain the alanine glycine-proline region [31]. The glycine-proline hinge region in the middle of the helical structure can benefit the pore-forming peptides [40]. Conversely, another study about cecropin attempting to delete the proline-glycine hinge resulted in decreasing antimicrobial activity, although there was an increase in anti-inflammatory activity [41].

Further, the net charge of the peptides has a significant impact on the activity of the peptides upon the deletion of three amino acid residues. Specifically, the deletion of negatively charged amino acid residues increased the net charge of the peptides as well as the antimicrobial activity of the peptides. Furthermore, the deletion of amino acid residues helped arrange positioning of hydrophobic amino acids in their structure. It increased the mean hydrophobicity of the peptides from 0.110 in CeHS-1 to 0.463 and 0.468 in CeHS-1 GP and CeHS-1 GPK, respectively. Although the mean hydrophobicity of the potent antimicrobial peptides varied, most of them had the mean hydrophobicity around this value [41,42,43].

Meanwhile, amphipathicity is the crucial key of antimicrobial activity of the peptides [44]. Amphipathicity of antimicrobial peptides structure reflects the proportion of hydrophobic and hydrophilic domains. Furthermore, it determines their polarization [16]. In α-helix structure, amphipathicity helps maintain the structure and increases binding to the membrane interface due to the formation of intramolecular hydrogen bonds [44,45]. In this study, the structure modification significantly increased the helicity of the peptides. As increasing their helicity, CeHS-1 GP and CeHS-1 GPK also performed the elevation on their antimicrobial activity compared to CeHS-1. The result was in line with the estimate that most helicity improvements usually increase the activity of peptides [46]. It was suspected that the addition of the glycine-proline hinge region increased the overall helicity value of the peptides [40]. Despite, the insertion of proline might be interrupted the helical span [47], the deletion of negative charge residues helped to conform an idealized amphipathic helix, with distinct hydrophobic and hydrophilic facets [22].

The modification of CeHS-1 GP to CeHS-1 GPK by Lysine substitution on Position 19 increased the positive charge and the hydrophobic moment of the peptides. In this case, the in silico analyzing result corresponded with another study [48]. In this study, the substitution had increased the ability of the peptides to disrupt the bacterial cell membrane in both the inner and outer membrane. The substitution played a crucial role in the disruption activity due to the efficient insertion of positive charge in the side chain of the peptides [15]. This charge strengthened the interaction with the anionic membrane by electrostatic interaction [46].

Like most antimicrobial peptides [18], the killing mechanism of these peptides is through membrane disruption activities. The sequences modification in CeHS-1 GP and CeHS-1 GPK increased the ability of the peptides to disrupt both the inner and outer membrane. It is shown in Figure 5, Figure 6 and Figure 7 that the mutant peptides disrupted the membrane more than the parent peptide. In this case, CeHS-1 GPK was disrupted more than CeHS-1 GP, corresponding with their ability to kill the bacteria. By increasing peptides concentration, the membrane disruption of CeHS-1 GP and CeHS-1 GPK escalated, suggested that membrane disruption activity of peptides worked through the carpet model. In this model, the accumulation of peptides at adequate concentration increased the curvature of the membrane, resulting in an increase in the number of spiral pores in the membrane [9]. The SEM analysis strengthened the assumption that the membrane disruption occurred through the carpet model. Figure 8c,d showed that the peptides treatment changed the integrity of the cell membrane. In the carpet model, the peptide micelles first touch the membrane and cover a small area, followed by the production of the pore and the hole in the cell membrane. To break the membrane into micelles, the peptides bind parallel to the membrane [24].

Meanwhile, the synergistic evaluation between CeHS-1 GPK and conventional antibiotics showed that the combinations with kanamycin have better synergism compared to ampicillin. Antibiotics synergism can be caused as a result of a relatively simple uptake effect. It can occur when one drug increases the permeability of the bacterial cell membrane to another drug [49]. As an antibiotic that combats the bacteria by inhibiting the 30s subunit in the protein synthesis [50], the uptake of kanamycin in the bacteria might be facilitated by the increase of membrane permeability by CeHS-1 GPK. On the other hand, the combination between CeHS-1 GPK and ampicillin did not show an increasing effect. As ampicillin targets cell wall synthesis while CeHS-1 GPK targets cell membrane, the combination of both can suppress ampicillin activity. In addition, this result aligns with another study focusing on a similar combination between vancomycin and colistin, where colistin triggered gene expression change in the bacteria such as those in the vancomycin-resistant mutant [51].

All in all, antimicrobial activity is not the only criterion for determining the quality of antimicrobial peptides for clinical purposes. Significant limiting factors, including hemolytic activity and cytotoxicity to mammalian cells, needed further identification [46]. The mutant showed increasing activity without elevating hemolytic activity. Yet, the cytotoxic test showed the increased cytotoxicity on CeHS-1 GPK was in line with the increasing of the concentration and antibacterial activity of this peptide. However, in the CeHS-1 GPK MIC value of 8 μg/mL, the cell viability was more than 80%. Likewise, CeHS-1 GP had low toxicity in its MIC value, which could still maintain cell viability over 80%. It indicated that both mutant peptides were harmless to the normal cells [52].

## 4. Materials and Methods

### 4.1. Peptides Design

The first 36 amino acid sequences of Heteroscorpine-1 (CeHS-1) were picked and analyzed by The Antimicrobial Peptide Database Program available online https://wangapd3.com/main.php, accessed on 17 January 2021 [23]. This program showed the sequence homology between CeHS-1 and the established potent antimicrobial peptides. Based on this analysis, CeHS-1 was used as the template for designing the new peptides, while the established antimicrobial peptides were used as the model for the ideal antimicrobial peptide structure. Upon the sequence analysis, the net charge, the mean hydrophobicity, and the mean amphipathic moment were calculated using the HeliQuest service at http://heliquest.ipmc.cnrs.fr/cgi-bin/ComputParams.py, accessed on 3 February 2021 [53]. Besides, this program was used to determine the two-dimensional structure of the mutants to optimize the positioning of the positive charge of the mutant. Meanwhile, the three-dimensional structure of the peptides was analyzed by the SWISS-MODEL program at https://swissmodel.expansy.org, accessed on 4 February 2021 [54].

### 4.2. Peptides Synthesis

All peptides were chemically synthesized. The purity of peptides was analyzed by high performance liquid chromatography (HPLC) (GenScript, Piscataway, NJ, USA), and all peptides were obtained at a purity grade more than 85% (Appendix A). Later, the molecular weight of synthetic peptides was analyzed by mass spectrophotometry (MS) to be compared with its theoretical molecular weight (Appendix A).

### 4.3. CD Spectroscopy

The secondary structures of the peptides were examined by CD spectroscopy. All the measurements were performed on a Jasco J-715 spectropolarimeter (Jasco, Easton, MD, USA). The CD spectra at 60 µg/mL peptide concentration in two different environments which are water and 40% trifluoroethanol aqueous solution (TFE) were recorded the averaged of six scans from 260 nm to 190 nm, using a quartz cell of 1 mm optical path length at room temperature. Data were collected at a scan speed of 50 nm/min, bandwidth of 1.0 nm, 1 s response, and 0.1 nm resolution [43]. The observed ellipticity, *θ* (mdeg), was converted to molar ellipticity [*θ*] (deg cm^2^/dmol) using the following equation [36]:(1)$$\left[\theta \right]={\left[\theta \right]}_{obs}\;\left(\frac{MRW}{10cd}\right)$$

Based on the equation, ${\left[\theta \right]}_{\mathrm{obs}}$ is the experimentally measured ellipticity, *MRW* is the mean residue molecular weight of the peptide (molecular weight divided by the number of peptide residue), *c* is the peptide concentration in mg/mL, and *d* is the path length of the cell in cm.

### 4.4. Antimicrobial Activity

The MICs of the peptides were determined in triplicate using the liquid microdilution method in the sterile 96-well microplate, followed by the protocol with some modification [55]. Briefly, tested bacteria were inoculated to 5 mL of nutrient broth (HiMedia Laboratories, India) and later cultured overnight at 37 °C with the shaking 180 rpm. The overnight cultures were recultured into the nutrient broth and incubated in the same condition. After the OD_600_ reached approximately 0.6, the cultures were diluted until the final concentration reached approximately 5 × 10^5^ CFU/mL and used for tested bacteria.

Meanwhile, 100 µL of two-fold serial dilutions of tested peptides and antibiotics in nutrient broth were prepared in the wells. Subsequently, 5 µL of tested bacteria was added to the wells and incubated at 37 °C for 16–24 h. Upon the incubation, the OD_600_ of each well was determined. The lowest concentration that could inhibit bacterial growth was defined as MIC. Meanwhile, in determining the MBC of the peptides, the amount of 25 µL culture with no bacterial growth was spread to nutrient agar and incubated at 37 °C for 16–24 h. As a result, the lowest concentration with no bacterial growth in the plate was defined as MBC [56].

### 4.5. Synergistic Activity Assay

The studies of synergistic activity between antibiotics and the peptides were performed with checkerboard titration methods. The peptides and antibiotics were set in several concentrations, and the tests were performed in triplicate. The synergistic activities were determined by the *FIC* index, calculated according to the equation:(2)$$FIC=FIC_{A}+FIC_{B}=\frac{A}{MIC_{A}}+\frac{B}{MIC_{B}}$$

While *A* and *B* are the MICs of Drugs *A* and *B* in the combination, *MIC_A_* and *MIC_B_* are the MICs of Drugs *A* and *B*, respectively. Besides, *FIC_A_* and *FIC_B_* are the FICs of Drug *A* and Drug *B*, respectively. The *FIC* index was interpreted as follows: ≤0.5 is synergy; 0.5 to 1.0 is addition; 1.0 to 4.0 is indifference; and ≥4.0 is antagonism [57].

### 4.6. Time Killing Assay

The tested bacterial strains were inoculated in nutrient broth until the log phase and were diluted until the cell concentration was approximately 5 × 10^6^ CFU/mL. Furthermore, the peptides were added and incubated at 37 °C for 24 h. At the particular time intervals of 0, 2, 4, 8, and 12 h, 25 µL of cultures was removed from the tube and plated into the nutrient agar plate. The plate was later incubated at 37 °C for 16–24 h, and the total number of colonies was determined [58].

### 4.7. Outer Membrane Permeability Assay

The permeability of the bacterial outer membrane treated with the peptide was analyzed using the fluorescent probe 1-N-phenyl naphthylamine (NPN) (Sigma–Aldrich, Steinheim, Germany) as previously described [38]. Briefly, a colony of the bacterial test was inoculated to 5 mL of nutrient broth and incubated overnight with shaking at 180 rpm. The amount of 50 µL overnight culture was recultured until the OD_600_ reached 0.5. Moreover, the cell was centrifuged at 10.000 rpm for 2 min, washed with HEPES buffer supplemented with 20 mM glucose pH 7.4, and resuspended with the same buffer to OD_600_ = 0.5. Subsequently, 50 μL of bacterial cells and 50 μL of HEPES buffer contained 20 μM NPN were mixed in the 96 well optical-bottom black plates. The peptides were added to the wells, and the fluorescence was immediately monitored at an excitation wavelength of 350 nm and an emission wavelength of 420 nm for 10 min at 30 sec intervals by EnSight Multimode Plate Reader (PerkinElmer, Waltham, MA, USA). The NPN uptake was later calculated using the equation, as follow:(3)$$NPN\;uptake=\left(F_{obs}-F_{B}\right)-\left(F_{control}-F_{B}\right)$$

Based on the equation above, *F_obs_* was the fluorescence of bacteria treated with peptides. At the same time, *F_control_* was the fluorescence of bacteria without treatment, while *F_B_* was the fluorescence of NPN in the media without bacteria.

### 4.8. Inner Membrane Permeability Assay

The bacterial cell was cultured overnight in nutrient broth, washed, and resuspended with HEPES buffer supplemented with 20 mM glucose pH 7.4 to 0.5 of OD_600_. The bacteria were mixed with propidium iodide (PI) (Sigma–Aldrich, Steinheim, Germany) to the final concentration of 10 µg/mL, and 100 µL of the mixing was added to the 96 well optical-bottom black plates. Afterward, peptides with different concentrations were added to the well, and the fluorescent was immediately measured at an excitation wavelength of 580 nm and an emission wavelength of 620 nm [25].

### 4.9. Scanning Electron Microscope

The bacterial cultures test was conducted in nutrient broth to log phase and centrifuged at 3000× *g* for 5 min. The cell pellet of the bacteria was washed twice with 10 mM sodium phosphate buffer, pH 7.2, and resuspended with the same buffer to a final concentration of 1 × 10^8^ CFU/mL (OD600 = 0.1). The aliquot of bacterial suspension was incubated with the half MIC of the peptides at 37 °C for 24 h. The tested bacteria were subsequently washed using PBS, fixed with 4% glutaraldehyde, and dehydrated by rinsing consequently with a series of ethanol solutions. Furthermore, the bacterial samples were individually coated with gold-palladium. The antimicrobial effect on each coated specimen was monitored under the scanning electron microscope (LEO Electron Mycroscopy, Cambridge, UK) operating at 12–20 kV. The bacterial cell untreated with peptides was used as a control.

### 4.10. Hemolytic Activity Assay

The hemolytic activity of the peptides was determined based on the hemolysis of human red blood cells (hRBCs). Briefly, 4% of hRBCs suspension was prepared in phosphate buffer saline (PBS), pH 7.4. A 100 µL suspension was transferred to a micro-centrifuge tube and added with 10 µL of peptides sample. The tube was later incubated at 37 °C for 1 h, followed by a centrifuge at 100× *g* for 5 min. The supernatant from each tube was withdrawn and moved into a 96-well plate for measurement at 415 nm. Upon the measurement, the hemolytic activity of the peptide was evaluated according to the equation, as follows:(4)$$Hemolysis\;\left(\%\right)=\left(\frac{S}{P}\right)\times 100$$

In the equation above, *S* is the absorbance of the peptides or negative control at 415 nm, while *P* is the absorbance of the positive control at 415 nm. Furthermore, the peptide solvent was used as a negative control, and 0.1% Triton X-100 was used as a positive control.

### 4.11. Cytotoxic Assay

The cytotoxicity of the peptides on the normal mammalian cells was performed into macrophage (RAW 264.7) cell line and human skin keratinocyte (HaCat) cell line using 3-[4,5-dimethylthiazol-2-yl]-2,5 diphenyltetrazolium bromide (MTT)-based cytotoxicity assay [42]. The cell was cultured in a DMEM culture medium, supplemented with 10% fetal bovine serum in a humidified atmosphere at 37 °C and 5% CO_2_. Upon the incubation, the cells were harvested and seeded to 96 well plates at the density of 2.5 × 10^4^ and 1 × 10^4^ cells per well for macrophage and HaCat cell lines, respectively. After 24 h of incubation, the cells were treated with the peptides in the different concentrations, followed by incubation for 24 h with the same condition. The medium was later replaced with 50 µL of 0.5 mg/mL MTT solution in PBS. The cell was re-incubated for 4 h until a formazan crystal was formed. Afterward, 100 µL DMSO was added, and the absorbance immediately was read in 570 nm. The cell viability is determined as follows:(5)$$\mathrm{Cell}\;\mathrm{viability}=\frac{Abs_{Sample}}{Abs_{control}}\times 100\%$$

As a note, the peptide solvent was used as a negative control comparison.

## 5. Conclusions

In conclusion, the researcher investigated whether the mutation of CeHS-1 through the addition of glycine-proline hinge region increased the antimicrobial activity of the peptides without the significant increase in toxicity. This study confirmed that CeHS-1 GP and CeHS-1 GPK showed potent activity to kill Gram-positive and Gram-negative bacteria through their ability to disrupt bacterial membrane cells. Furthermore, the mutant peptides also showed low toxicity against normal mammalian cells.

## Figures and Tables

**Figure 1 molecules-26-05872-f001:**
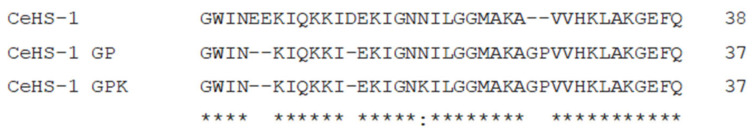
The comparison between the amino acid sequences of CeHS-1 and its analogs. (*) indicate identical amino acids. (:) indicate the similar amino acid.

**Figure 2 molecules-26-05872-f002:**
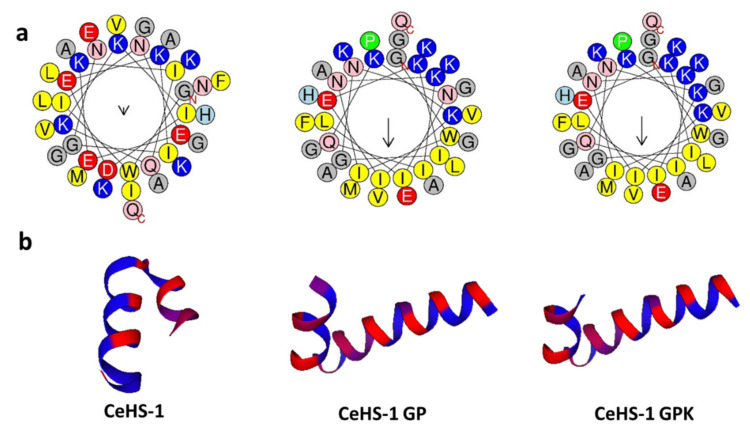
Structural projection of peptides: (**a**) The Helical wheel projection of peptides, (**b**) theoretical tridimensional projection of peptides.

**Figure 3 molecules-26-05872-f003:**
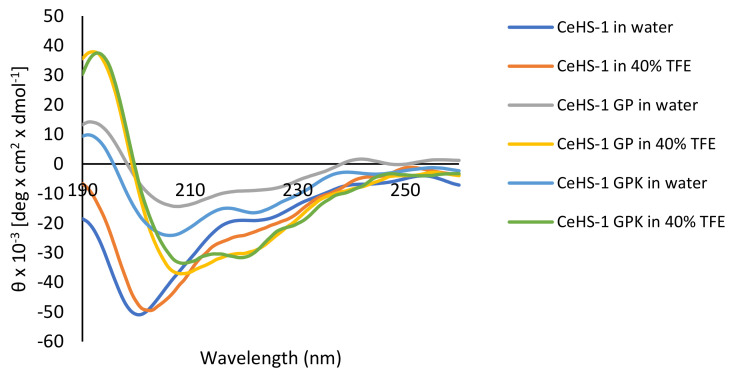
Secondary structures of the peptides determined by circular dichroism spectroscopy.

**Figure 4 molecules-26-05872-f004:**
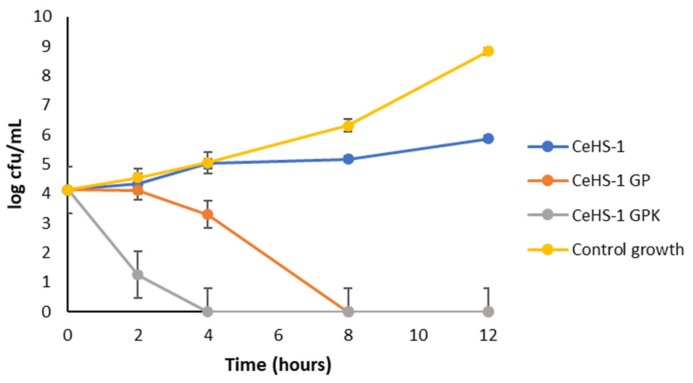
Time killing of peptides.

**Figure 5 molecules-26-05872-f005:**
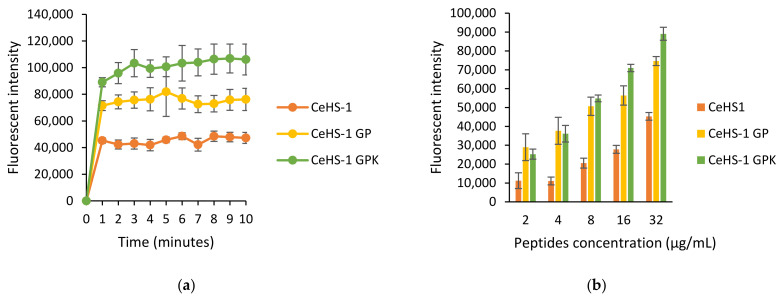
The effect of peptides on outer membrane of *E. coli*: (**a**) the effects after 10 min treatment; (**b**) the effects in the various concentration of peptides.

**Figure 6 molecules-26-05872-f006:**
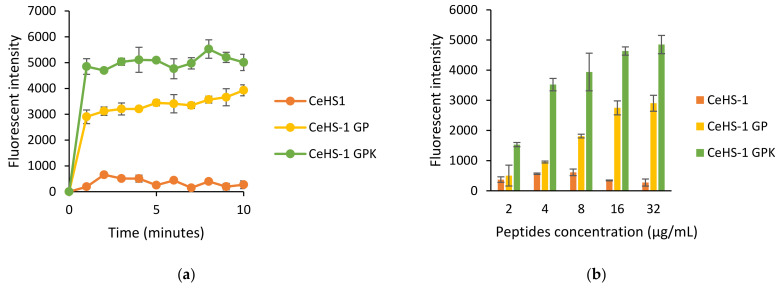
The effect of peptides on inner membrane of *E. coli*: (**a**) the effects after 10 min treatment; (**b**) the effects in the various concentration of peptides.

**Figure 7 molecules-26-05872-f007:**
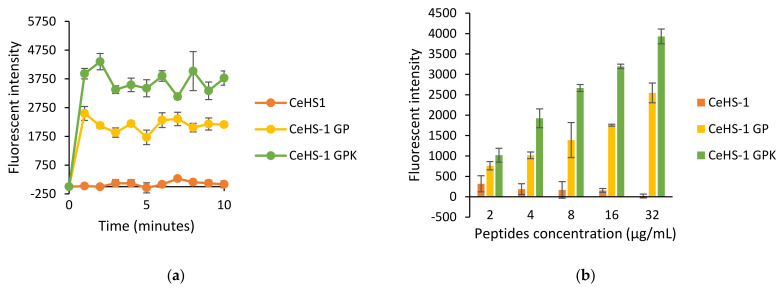
The effect of peptides on inner membrane of *S. aureus*: (**a**) the effects after 10 min treatment; (**b**) the effects in the various concentration of peptides.

**Figure 8 molecules-26-05872-f008:**
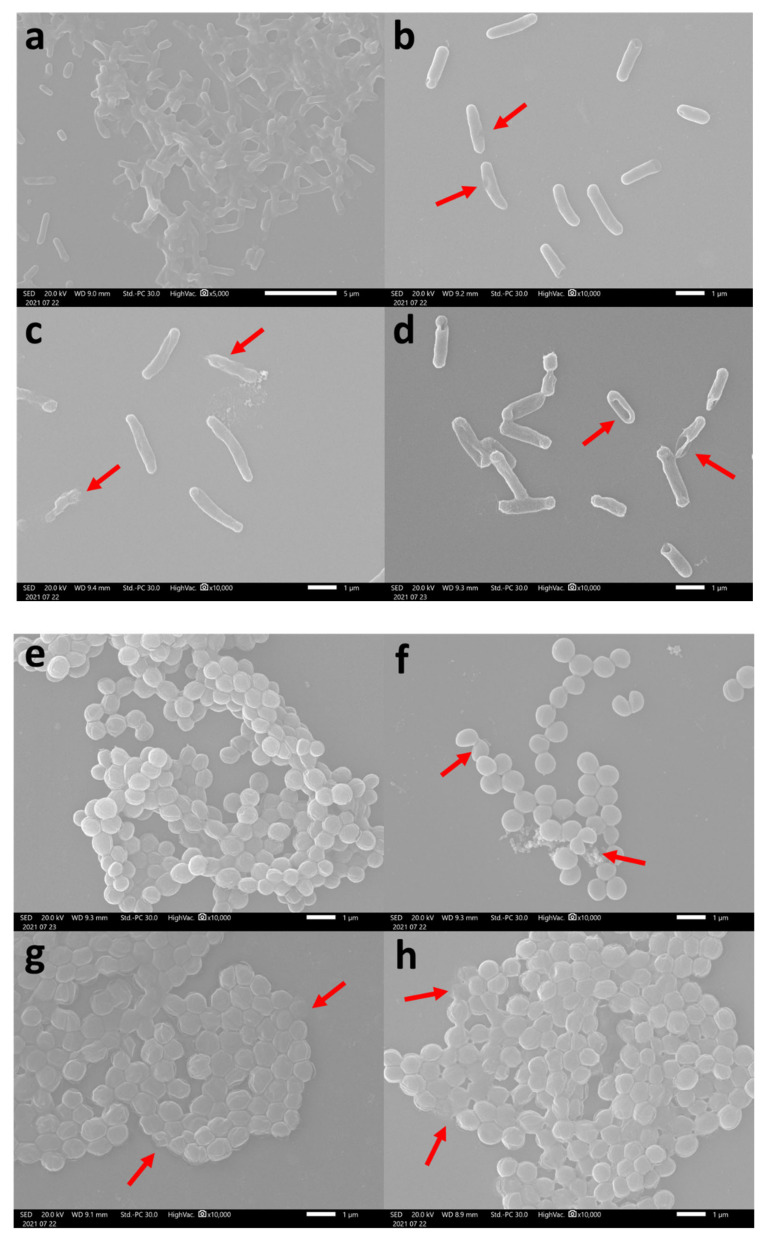
SEM image of bacteria treated with peptides: (**a**) *E. coli* without treatment; (**b**). *E. coli* treated with CeHS1; (**c**). *E. coli* treated with CeHS-1 GP; (**d**). *E. coli* treated with CeHS-1 GPK; (**e**). *S. aureus* without treatment; (**f**). *S. aureus* treated with CeHS1; (**g**). *S. aureus* treated with CeHS-1 GP; (**h**). *S. aureus* treated with CeHS-1 GPK.

**Figure 9 molecules-26-05872-f009:**
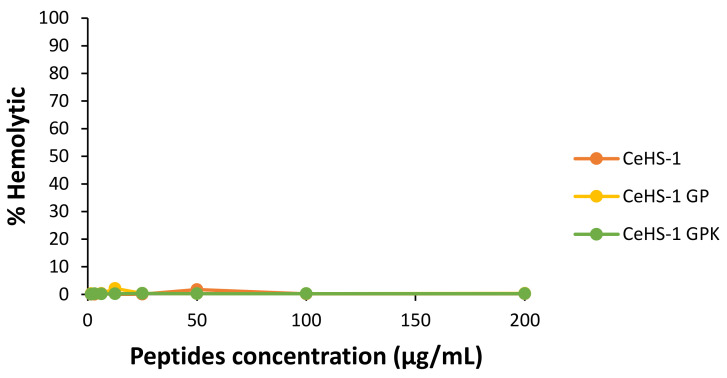
Hemolytic activity of peptides against human red blood cells after 1 h exposure.

**Figure 10 molecules-26-05872-f010:**
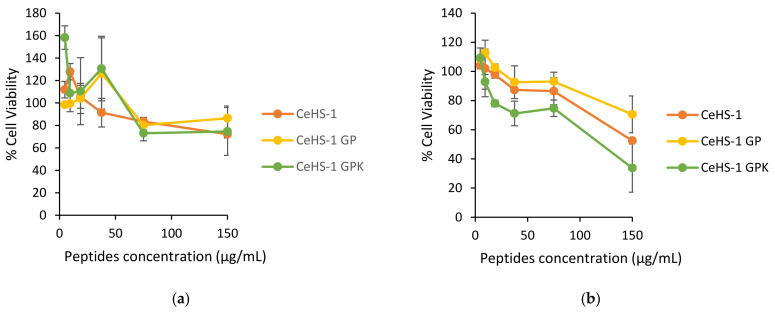
Cytotoxic activity of peptides in various concentration: (**a**) towards HaCat cell line; (**b**) towards a macrophage cell line.

**Table 1 molecules-26-05872-t001:** Alignment of the sequence of CeHS-1 with the established antimicrobial peptides.

Peptides	UniProtKB ID	Origin	Sequence	AA	Identity (%)	Reference
CeHS-1	-	*Heterometrus laoticus*	GWINEEKIQKKIDEKIGNNILGGMA KA--VVHKLAKGEFQ--	38	-	This work
Cecropin B	AP00128	*Antheraea pernyi*	KW----KIFKKI-EKVGRNIRNGII KAGPAVAVL--GEAKAL	35	42.85	[30]
CM4	AP01259	*Bombyx mori*	RWKIFKKI-----EKVGQNIRDGIV KAGPAVAVVGQAATI	35	41.86	[33]
Hyphancin IIIF	AP00348	*Hyphantria cunea*	RWKVFKKI-----EKVGRNIRDGVI KAGPAIAVVGQAKAL	35	41.86	[34]

The sequence alignment was obtained from the Antimicrobial Peptides Database (online at https://wangapd3.com/main.php, accessed on 17 January 2021). AA is the number of amino acids; Identity (%) is sequence identity in percentage.

**Table 2 molecules-26-05872-t002:** The Physicochemical Prediction of the Peptides.

Peptides	Molecular Weight (MW)		Charge	%H	H	μH	α-Helix
	Theoretical	Observed					
CeHS-1	4235.93	4234.8	+2	39.47	0.228	0.110	76.32%
CeHS-1 GP	4016.78	4016.0	+5	43.24	0.309	0.463	70.27%
CeHS-1 GPK	4030.85	4029.5	+6	43.24	0.299	0.468	67.57%

%H: Hydrophobic residues, H: hydrophobicity, μH: hydrophobic moment, MW was calculated by PepCalc.com (https://pepcalc.com/, accessed on 3 February 2021) and observed by mass spectrophotometry. The charge, hydrophobicity, and hydrophobic moment were calculated using Heliquest (https://heliquest.ipmc.cnrs.fr/cgi-bin/ComputParams.py, accessed on 3 February 2021), and the α-helix propensity was predicted by HNN secondary structure prediction method https://npsa-prabi.ibcp.fr/cgi-bin/, accesses on 4 February 2021.

**Table 3 molecules-26-05872-t003:** The Determination of MIC of CeHS-1, CeHS-1 GP, and CeHS-1 GPK.

Peptides	% of Content in Water		% of Content in 40% TFE	
	α-Helix	β-Sheet	α-Helix	β-Sheet
CeHS-1	8.02	22.14	8.02	22.14
CeHS-1 GP	62.6	3.79	87.59	0.48
CeHS-1 GPK	62.6	3.79	87.59	0.48

The percentage content in α-helix and β-sheet of the peptides was estimated by the K2D2 method (http://cbdm-01.zdv.uni-mainz.de/~andrade/k2d2/, accessed on 17 September 2021) with the estimated maximum error was >0.32.

**Table 4 molecules-26-05872-t004:** The Determination of MIC of CeHS-1, CeHS-1 GP, and CeHS-1 GPK.

Bacterial Strain	CeHS-1		CeHS-1 GP		CeHS-1 GPK		Ampicillin	
	MIC	MBC	MIC	MBC	MIC	MBC	MIC	MBC
*K. pneumoniae* ATCC 27853	64	>128	32	64	32	128	0.06	0.13
*E. coli* ATCC 25922	64	>128	16	16	8	8	1	1
*B. subtilis* TISR 008	64	64	32	64	8	16	0.06	0.06
*S. aureus* ATCC 25923	64	>128	32	128	16	64	0.06	0.25
*P. aeruginosa* ATCC 27853	64	64	32	32	16	32	0.13	0.25
resistant *K. pneumoniae*	16	>128	32	>128	32	>128	>128	>128
*MRSA*	128	>128	128	>128	128	>128	32	32

MIC: minimum inhibitory concentration; MBC: minimum bactericidal concentration; MIC and MBC values are determined in μg/mL.

**Table 5 molecules-26-05872-t005:** The FIC of Synergistic Activity between CeHS-1 GPK and Antibiotics.

Bacterial Strain	FIC in Combination with	
	Ampicillin	Kanamycin
*E. coli* ATCC 25922	0.75	1
*B. subtilis* TISR 008	0.75	0.5
*S. aureus* ATCC 25923	3	2
*P. aeruginosa* ATCC 27853	0.75	0.5

TFIC is the index interpreting the combination effect. The value is meaning as follows: ≤0.5 is synergy; 0.5 to 1.0 is addition; 1.0 to 4.0 is indifference; and ≥4.0 is antagonism.

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
