# Peer review of "Novel Antimicrobial Peptides from a Cecropin-Like Region of Heteroscorpine-1 from Heterometrus laoticus Venom with Membrane Disruption Activity"

_molecules, 2021, doi:10.3390/molecules26195872_

Round 1

Reviewer 1 Report

MOLECULES Manuscript: 1391944
Title: Novel Antimicrobial Peptides from a Cecropin-like Region of Heteroscorpine-1 from Heterometrus laoticus Venom with Membrane Disruption Activity
Authors: Rima Erviana et al.

This manuscript describes the design, chemical synthesis and use of peptides with similar structure as that of Heteroscorpine-1 N-terminal segment, which has similarities to the amino acid sequence of cecropin. After synthesis and partial purification, the peptides were shown to have an antimicrobial activity against Gram-positive and Gram-negative microorganisms.
The manuscript reads well and contains some novel data, although the strategy used follows known properties of this type of peptides: hydrophobicity, net charge and a particular secondary structure. The authors have deleted negatively charges amino acids, added positively charged ones and directed the positions of hydrophobic amino acid residues along the sequence. In this respect the results are somehow confirmatory of what was already known in the literature.
The initial model was taken from Heteroscorpine-1, which is a peptide purified from the venom of the scorpion Heterometrus laoticus. The peptide belongs to the family of the scorpines. The original scorpine was purified from Pandinus imperator venom (Conde et al., FEBS Letters 471:165-168 (2000). It is a complex peptide containing two structurally different segments. The N-terminal amino acid is similar to the antimicrobial cecropin and the C-terminal to defensins or to K+-channel blocking peptides (Diego-Garcia et al. Cellular and Molecular Life Sciences 65:187-200 (2008).
In my opinion the manuscript can be improved by some minor corrections.
1. The chemical synthesis of the modified peptides was done in the laboratory or purchased from a commercial source? For future publications dealing with this subject, it would be important to show (even if just supplementary material) the purification used (HPLC profile). An 85% purity in my opinion is not sufficient, specially if the idea is to use it eventually for clinical trials.
2. In the same directions, for future work, I would suggest a confirmation of the alpha-helix formation of the synthesized peptides. This can be easily made by measurements using a circular dichroism apparatus. The correct helical format can be added by the addition of TFE (trifluoroacetic acid), which is known to facilitate helix formation.
3. Please define abbreviations (FIC, MIC, MRSA, MBC, etc.) in the main text first time mentioned. This would avoid the need for the readers to search into the material and methods the corresponding definitions.
4. Need to homogenized nomenclature: Fig.2A, Fig2B, instead of Fig2a and Fig2b. Idem with Fig.7 (small or capital letters)?

Author Response

Reviewer’s Comments

This manuscript describes the design, chemical synthesis and use of peptides with similar structure as that of Heteroscorpine-1 N-terminal segment, which has similarities to the amino acid sequence of cecropin. After synthesis and partial purification, the peptides were shown to have an antimicrobial activity against Gram-positive and Gram-negative microorganisms.

The manuscript reads well and contains some novel data, although the strategy used follows known properties of this type of peptides: hydrophobicity, net charge and a particular secondary structure. The authors have deleted negatively charges amino acids, added positively charged ones and directed the positions of hydrophobic amino acid residues along the sequence. In this respect the results are somehow confirmatory of what was already known in the literature.

The initial model was taken from Heteroscorpine-1, which is a peptide purified from the venom of the scorpion Heterometrus laoticus. The peptide belongs to the family of the scorpines. The original scorpine was purified from Pandinus imperator venom (Conde et al., FEBS Letters 471:165-168 (2000). It is a complex peptide containing two structurally different segments. The N-terminal amino acid is similar to the antimicrobial cecropin and the C-terminal to defensins or to K+-channel blocking peptides (Diego-Garcia et al. Cellular and Molecular Life Sciences 65:187-200 (2008).

In my opinion the manuscript can be improved by some minor corrections.

Answer

Dear Respected Reviewer,

First of all, I am so happy to have your comments and suggestion. It is so encouraging and helping us to revise our paper to be better. Therefore, I am so appreciated with your response, comment, and suggestions. As your suggestions, I do agree that this manuscript should be revised and improved.

Therefore, I have tried to answer and follow your question, comment, and suggestions. In terms of specific issues. I have added some discussion about Heteroscorpine-1, which belongs to the family of the scorpine. I have also revised and informed more details on some in-depth information about the structure characteristic of the scorpine family and its bioactivity in the introduction. As well, I have cited your suggested reference.

In some details of your comments, please have a look at my answers and my main manuscript. Because of your comments, I am sure that my paper it is much better and clearer now. Therefore, I hope my paper can pass and meet your requirements to be published in this journal.

Once again, many thanks indeed for your excellent review, and waiting forward to hearing from you.

Best Regards

Sakda Daduang

Here is some answer from your specific opinion on the manuscript:

  1. The chemical synthesis of the modified peptides was done in the laboratory or purchased from a commercial source? For future publications dealing with this subject, it would be important to show (even if just supplementary material) the purification used (HPLC profile). An 85% purity in my opinion is not sufficient, specially if the idea is to use it eventually for clinical trials.

Answer:

The chemical synthesis of all peptides, including CeHS-1, CeHS-1 GP, and CeHS-1 GPK was purchased from a commercial source (Genscript USA, Inc.). In this manuscript, I have added some information about the HPLC and MS analysis of peptides as the supplementary material. Regarding the purity grade of the peptides, in our opinion, for the current study, 85% purity is still appropriate for comparing the activity between nature and modified peptides in vitro. Another study about antimicrobial peptides with similar purity produced an acceptable result (DOI: 10.1016/j.gene.2018.05.106).  However, I do really agree with your opinion that in the future studies, especially in the clinical trial, higher purity should be an important issue to consider.

  1. In the same directions, for future work, I would suggest a confirmation of the alpha-helix formation of the synthesized peptides. This can be easily made by measurements using a circular dichroism apparatus. The correct helical format can be added by the addition of TFE (trifluoroacetic acid), which is known to facilitate helix formation.

Answer:

I have revised and already added the information about the secondary structure determination of the peptides by measure the circular dichroism spectra and discussed the structure in the manuscript. To determine their structure in aqueous and membrane mimic environment, water and 40% TFE was used as the peptides solvent. Please have a look at our manuscript at the result section 2.2.

  1. Please define abbreviations (FIC, MIC, MRSA, MBC, etc.) in the main text first time mentioned. This would avoid the need for the readers to search into the material and methods the corresponding definitions.

Answer:

I do agree and already revised it. For this issue, I have already checked all the abbreviation and defined in the first time they were mentioned. They include MIC, MBC, MRSA, FIC, NPN, PI, HPLC, etc. Please have a look at my revised manuscript.

  1. Need to homogenized nomenclature: Fig.2A, Fig2B, instead of Fig2a and Fig2b. Idem with Fig.7 (small or capital letters)?

Answer:

I do agree and revised it, so it is much clearer now. For instance, the nomenclature of the figure was already homogenized with other pictures, which used small letters. We changed the letter in figure 2 and figure 7 to the small letters.

Thank you so much for your constructive comments.

Reviewer 2 Report

The submitted manuscript “Novel Antimicrobial Peptides from a Cecropin-like Region of Heteroscorpine-1 from Heterometrus laoticus Venom with Membrane Disruption Activity” by Erviana et al developed two modified antimicrobial peptides, CeHS-1 GP and CeHS-1 GPK, derived from heteroscorpine-1 (CeHS-1). The further bioactivity and mode of action showed their potency against target bacteria. It is an interesting study. However, there is a lack of circular dichroism determination of the two peptides, as well as the no conventional antibiotics control for bioactivity testing. At this stage, I recommend this manuscript be published after major revision.

There are few minor comments,

  1. There is no characterisation of the synthetic peptides, such as RP-HPLC and mass spec.
  2. Page 2 line 48, cephalosporin, a β-lactam antibiotic, has been used for conjugate with peptide for synergy study, which should be discussed in the manuscript such as Peptide Science. 2018;110:e24059, DOI:1002/pep2.24059; RSC Adv., 2012, 2, 2480-2492, DOI: 10.1039/C2RA01351G.
  3. Page 2 line 76, they mentioned the database of 5000 recorded antimicrobial peptides. A reference, such as Nucleic Acids Research 44, D1087-D1093, should be referred.
  4. Fig 4, 5 and 6, the authors should be clear of the time point analysed for Fig b.
  5. The quality of fig 7 is too poor to judge their claim.

Author Response

Reviewer’s Comments

 The submitted manuscript “Novel Antimicrobial Peptides from a Cecropin-like Region of Heteroscorpine-1 from Heterometrus laoticus Venom with Membrane Disruption Activity” by Erviana et al developed two modified antimicrobial peptides, CeHS-1 GP and CeHS-1 GPK, derived from heteroscorpine-1 (CeHS-1). The further bioactivity and mode of action showed their potency against target bacteria. It is an interesting study. However, there is a lack of circular dichroism determination of the two peptides, as well as the no conventional antibiotics control for bioactivity testing. At this stage, I recommend this manuscript be published after major revision.

Answer:

Dear respected Reviewer,

I am so pleased to have your supportive comments, opportunity, and acceptance chance. It is so encouraging and supporting us to improve our manuscript. I do really agree with your suggestion that circular dichroism determination is important to investigate the secondary structure of the peptides. Therefore, I have revised and added some data about the secondary structure determination of the peptides by measuring the circular dichroism spectra and discussed the secondary structure of peptides in the manuscript. To determine their structure in aqueous and membrane mimic environment, water and 40% TFE were used as the solvent. You might have a look at my result (2.2), method (4.3), and discussion.

In the bioactivity testing, I have also added some data to compare the antimicrobial activity of the peptides and the conventional antibiotic (Ampicillin). We compared their MIC and MBC value, as shown in table 4. For more detailed information, could you please check my manuscript.

Overall, I have done and followed your suggestion accordingly. I have improved it in terms of the language, so this paper is much clearer and less grammatical errors.

I hope this revision is sufficient and meets your requirements now.

Many thanks for your support and help.

Best regards,

Sakda Daduang

In some details of my revision and response to your minor suggestion, you might read my response here.

  • There is no characterisation of the synthetic peptides, such as RP-HPLC and mass spec.

Answer:

I have revised and already added some information about the characterization of the synthetic peptides. These are the HPLC analysis contained in figure supplement 1-3 and MS analysis contained in figure supplement 4-6.

  • Page 2 line 48, cephalosporin, a β-lactam antibiotic, has been used for conjugate with peptide for synergy study, which should be discussed in the manuscript such as Peptide Science. 2018;110:e24059, DOI:1002/pep2.24059; RSC Adv., 2012, 2, 2480-2492, DOI: 10.1039/C2RA01351G.

Answer:

I do agree, and I have added some discussion about the conjugated drugs in my manuscript. I also have put some references as your suggestion. Thank you.

  • Page 2 line 76, they mentioned the database of 5000 recorded antimicrobial peptides. A reference, such as Nucleic Acids Research 44, D1087-D1093, should be referred.

Answer:

I am pleased to have these suggested references. I have revised and put it in my manuscript. Thank you.

  • Fig 4, 5 and 6, the authors should be clear of the time point analysed for Fig b.

Answer:

I have revised and made it clearer now. We already added the time point data analysis for figure 4b, 5b, and 6b in the manuscript on page 8, line 252. The effect of the peptides exposure on the inner and outer membrane permeability was determined and measured every minute for 10 minutes. We took the data from the 2nd minute after exposure to compare the ability of the peptides to disrupt the membrane permeability in the various concentration of peptides.

  •  The quality of fig 7 is too poor to judge their claim.

Answer:

I do agree so to clarify figure 7 and make it clearer. I have revised and tried to provide a better picture with the highest quality. In addition, I have added some arrow to point the important part of the picture that show the activity of the peptides on bacterial cell membranes.

Once again, thank you for your supportive encouragement on this manuscript. Because of you, my manuscripts are much better and understandable now.

Round 2

Reviewer 2 Report

The authors have addressed my comments and I recommend the acceptance of the publication.